# COVID-19 Infection among Elite Football Players: A Nationwide Prospective Cohort Study

**DOI:** 10.3390/vaccines10050634

**Published:** 2022-04-19

**Authors:** Dimitrios Papagiannis, Theodoros Laios, Konstantinos Tryposkiadis, Konstantinos Kouriotis, Xenophon Roussis, Georgios Basdekis, Panagiotis Boudouris, Christos Cholevas, Stergios Karakitsios, Pindaros Kakavas, Theoharis Kiriakidis, Panagiotis Kouloumentas, Georgios Kouvidis, Grigoris Manoudis, Pantelis Nikolaou, Christos Theos, Andreas-Nikolaos Piskopakis, Ioannis Rallis, Stavros Ristanis, Alexandros Toliopoulos, Grigoris Zisis, Yiannis Theodorakis, Konstantinos I. Gourgoulianis, Georgios Rachiotis

**Affiliations:** 1Public Health & Vaccines Lab, Department of Nursing, School of Health Sciences, University of Thessaly Larissa, 411 10 Larissa, Greece; 2Super League Greece Mesogeion 174, 151 25 Maroussi, Greece; theodoros.laios@superleaguegreece.net (T.L.); kostas.kouriotis@superleaguegreece.net (K.K.); covid19@slgr.gr (X.R.); info@themedicalproject.gr (G.B.); info@asterastripolis.gr (P.B.); gpcholevas@gmail.com (C.C.); medicalrecords@metropolitan-hospital.gr (S.K.); pindaros.kakavas@gmail.com (P.K.); xaris220589@gmail.com (T.K.); info@kouloumentas.gr (P.K.); medicalofi@gmail.com (G.K.); info@volosnfc.gr (G.M.); nikp@otenet.gr (P.N.); christostheos@gmail.com (C.T.); piskopakis@gmail.com (A.-N.P.); arthroskopisi@gmail.com (I.R.); ristanis@gmail.com (S.R.); al.toliop@gmail.com (A.T.); info@panetolikos.gr (G.Z.); 3Institute of Applied Health Research, University of Birmingham, Birmingham B15 2TT, UK; kxt859@student.bham.ac.uk; 4Department of Physical Education and Sport Sciences, University of Thessaly, 382 21 Volos, Greece; theodorakis@uth.gr; 5Department of Respiratory Medicine, School of Health Sciences, University of Thessaly, 411 10 Larissa, Greece; kgourg@uth.gr; 6Department of Hygiene and Epidemiology, School of Health Sciences, Faculty of Medicine, University of Thessaly, 411 10 Larissa, Greece; grachiotis@gmail.com

**Keywords:** COVID-19, incidence, relative risk, professional footballers, transmission

## Abstract

Little is known about the risk of COVID-19 infection among footballers. We aimed to investigate the incidence and characteristics of COVID-19 infection among footballers. In total, 480 football players of Super League Greece and 420 staff members participated in a prospective cohort study, which took place from May 2020 to May 2021. Nasopharyngeal swabs were collected from footballers and staff members weekly. All samples (*n* = 43,975) collected were tested using the reverse transcriptase polymerase chain reaction (RT-PCR) test for the detection of “SARS-CoV-2”. In total, 190 positive cases (130 among professional football players and 60 among staff) were recorded. Out of the 190 cases that turned positive, 64 (34%) cases were considered as symptomatic, and 126 (66%) cases were asymptomatic. The incidence rate of a positive test result for footballers was 0.57% (confidence interval (CI) 0.48–0.68%) and for staff members it was 0.27% (CI 0.20%, 0.34%), respectively. Footballers recorded a twofold increased risk of COVID-19 infection in comparison to staff members (relative risk = 2.16; 95% CI = 1.59–2.93; *p*-value < 0.001). No significant transmission events were observed during the follow-up period. We found a low incidence of COVID-19 infection among professional footballers over a long follow-up period. Furthermore, the implementation of a weekly diagnostic testing (RT-PCR) was critical to break the transmission chain of COVID-19, especially among asymptomatic football players and staff members.

## 1. Introduction

The COVID-19 pandemic induced health risks and restrictions of opportunities for training and competition in elite football players. SARS-CoV-2 virus spreads via respiratory droplets primarily through close-range person-to-person contact and can lead to coronavirus disease 2019 (COVID-19). In addition, airborne transmission (indirect route of transmission) represents an additional (indirect) route of transmission. The proportion of asymptomatic infections reported in the literature varies widely depending on the study design and timing of COVID-19 testing [1,2]. A systematic review suggested that 40% of the COVID-19-infected individuals are asymptomatic, and three-quarters of persons who receive a positive PCR test result but have no symptoms at the time of testing will remain asymptomatic [3]. Following the eruption of the COVID-19 pandemic, the governments announced travel and health restrictions and imposed national lockdowns to safeguard society from the global pandemic. Consequently, football games were halted across the European continent. In this context, the Union of European Football Associations (UEFA) invited European football’s key stakeholders for a videoconference on 17 March 2020. Participants included representatives from all 55 UEFA member associations, the European Club Association (ECA), the European Leagues (EL), and the International Federation of Professional Footballers (FIFPRO Europe). Everyone attending the meeting committed to a united response to the pandemic that would prioritize the health and safety of players, staff, and officials [4]. The Greek Super League followed the medical regulatory of UEFA Hygiene Protocol for completing the season. The league participated according to the protocol throughout the games, regarding the training of the teams, and considering the special group training protocols as announced by the Special Health Committee and the General Association of Sports. The highest importance was given to respecting hygiene practices and health regulations to avoid COVID-19 spread. Seventy-two hours before the start of the games, all the players, the match officials, the coaching staff, and team staff need to be examined for COVID-19. The gold standard test to diagnose COVID-19 is the detection of SARS-CoV-2 RNA using reverse transcription polymerase chain reaction (RT-PCR) from a nasopharyngeal swab. The World Health Organization (WHO) in March 2020 recommended nasopharyngeal swab (NPS) samples for detection of SARS-CoV-2 and PCR testing of asymptomatic or mildly symptomatic contacts can be considered in the assessment of individuals who have had contact with a COVID-19 case [5]. Little is known about the incidence of COVID-19 infection among professional footballers. Although there are some lines of evidence suggesting a low risk of COVID-19 transmission and a mild illness (in terms of symptomatic status and need for hospitalization [6,7] among professional footballers, little is known about the relative risk of footballers for the acquisition of COVID-19 infection [8,9,10,11]. In addition, many of the relevant studies employed a short follow-up period. Consequently, the aim of our study was to investigate the incidence, relative risk, and characteristics of COVID-19 infection among high-level footballers over a period of 12 months.

## 2. Materials and Method

We performed a prospective cohort study with a longitudinal selection of nasopharyngeal and oropharyngeal samples in 14 high-level football teams in Greece. As part of a surveillance program of COVID-19 infection, all participants provided weekly nasal swabs appropriate to test for the detection of SARS-CoV-2. The results were sent weekly to the Hygiene Committee of Super League Hellas as part of the Super League occupational health and safety program for the surveillance of the COVID-19 pandemic. Participants were informed that data from the research protocol would be treated anonymously according to the principles of the Helsinki Declaration. Τhe protocol of the study was approved by the Institutional Review Board of Hygiene Committee of Super League Greece (1974/5-5-2020). This board acts as the official ethical body for the research projects related football players and staff of Super League Greece.

In total, 480 male football players of Super League Greece (mean age 27.45, SD ± 4.83 range 18–38) and 420 staff members (398 males 94.8% and 22 females 5.2%, range 31–68) were invited to participate in a prospective cohort study, which took place from May 2020 to May 2021. All football players and staff members of Super League Greece were recruited to take part in the study. The response rate of the participants in the present study was 100%. Staff members of a football club include different jobs involved in the backroom of a football club. The number of staff members depends on the size of the football club. The most usual jobs are the first coach of a team, the assistant manager, the goalkeeping coach, chief analyst, fitness coach, nutritionist, medical staff, physiotherapist, masseur, kit managers, press manager, and security manager.

### 2.1. Laboratory Testing

Nasopharyngeal swabs were collected from footballers and staff members weekly (Figure 1). The sampling process was performed by the members of the medical team of Super League Greece 1 football clubs. All samples collected were tested using reverse transcriptase polymerase chain reaction (RT-PCR) for the detection of SARS-CoV-2. RNA was extracted and determined by RT-PCR targeting the E, N, and RdRp genes of SARS-CoV-2, according to the WHO laboratory protocols [12]. The cycle threshold values of RT-PCR were used as qualitative indicators of viral load of SARS-CoV-2 RNA in specimens, with lower cycle threshold values corresponding to higher viral copy numbers.

### 2.2. History Taking and Physical Examinations

The physician of each football club obtained each footballer’s history with emphasis on COVID-19-related symptoms (e.g., fever, cough, runny nose, sore throat, smell, and test disorders).

### 2.3. Statistical Analysis

Data were tabulated and descriptive statistics were presented in terms of absolute (*n*) and relative (percentage; %) frequencies. For both athletes and staff members, bar charts were used in order to display the number of positive results over the total number of samples collected, per each of the 54 study weeks. The overall risk of a positive result for athletes $\hat{(R_{1}})$ and staff members $\hat{(R_{2}})$ was estimated as the total number of positive results divided by the total number of samples collected over the study time period, with a 95% confidence interval constructed using exact binomial methods [13]. If an athlete or a staff member was tested more than once, their results were included multiple times in the analysis. In order to compare the risk of a positive test results for athletes to that for staff members, a relative risk was in turn calculated as $\hat{RR}=\frac{\hat{R_{1}}}{\hat{R_{2}}}$. This estimate was presented along with a 95% confidence interval and p-value, which were calculated as described in Morris and Gardner [14] and Altman and Bland [15], respectively. All analyses were performed using Stata 17.0 [16].

## 3. Results

In the present study, 900 participants were invited by the medical officers of each team to participate in the Super League occupational health and safety program against COVID-19 pandemic, and 900 accepted the invitation (response rate: 100%). The participants were under surveillance over a period of 12 months (from May 2020 to May 2021). After examination of the 44,165 RT-PCR tests, we found 190 positive tests for SARS-CoV-2 RNA (incidence rate: 0.0043%). One hundred and thirty (130) cases were identified among football players (incidence rate: 0.0057%), and sixty cases were identified among staff (incidence rate: 0.0027%) (Table 1). Out of the 190 cases that turned positive, 64 cases were symptomatic (34%) and 126 (66%) were asymptomatic. An increased incidence rate was observed by the week 45 of 2020 (Figure 1). None of the players and staff testing positive or reactive at the study period required hospital admission or medical attention other than limited symptomatic treatment.

In bivariate analysis between football players and staff members the risk of a positive test result for athletes was 0.57% (CI 0.48%, 0.68%), and the risk of a positive test result for staff members was 0.27% (CI 0.20%, 0. 34%). The relative risk (athlete vs. staff) was 2.16 times higher risk for athletes compared to staff members (CI 1.59, 2.93), *p*-value < 0.001 (Table 1).

## 4. Discussion

In this prospective cohort study, we found an incidence rate of COVID-19 infection among Greek professional footballers at 0.57%. This rate is very similar to that reported by Pedersen et al. among Danish elite footballers. They reported a 0.53% rate of COVID-19 infection, and the positivity rate of testing was 0.06% during the observation period, and the players testing positive were asymptomatic at the time of testing [10]. Nevertheless, we employed a significantly longer follow-up period (12 months) in our study. Further, a two-month cohort study among footballers in German Bundesliga revealed a similar COVID-19 infection rate and very low risk of SARS-CoV-2 transmission during football match play [8]. Another German study found no evidence of transmission of SARS-CoV-2 on the pitch as verified by intensive PCR testing among professional footballers [9]. Notably, a prospective cohort study among players, staff, and referees of the national professional league of Qatar during a truncated football season of nine weeks reported a COVID-19 infection rate among footballers at 4.57%. In addition, contact tracing revealed that of the total number of players with positive or reactive PCR tests during the observation period, 55% of them did not know where or through whom they might have become infected [17]. All the above-mentioned data indicate a low risk of COVID-19 infection among professional footballers. It is well known that COVID-19 infection may be transmitted through direct contact related to large respiratory droplets and airborne mode of transmission (indirect mode of transmission related to droplet nuclei). Given that football is an outdoor activity the risk of airborne transmission can be considered as very low, or even negligible. With respect to transmission via large respiratory droplets (direct mode of transmission), there is evidence based on video records of football games that shows that direct contact between professional footballers in the field is rare in frequency and of short duration (<3–6 s) [9]. There is also some evidence that other outdoor sport activities (e.g., ski) are associated with a very low risk of COVID-19 transmission [18]. Another outdoor sport is rugby, which has repeated contact and close proximity interactions between players. According to Jones et al., despite the frequent interactions between SARS-CoV-2-positive players and other players, the data of their study suggest that SARS-CoV-2 transmission is limited during rugby league matches [19]. In the present study, we found 64 cases of COVID-19 illness with symptoms (33.68%), and the 126 cases of the laboratory confirmed infections were reported without symptoms (66.32%). Asymptomatic COVID-19 infections represent a crucial point in the prevention and control of COVID-19 pandemic. Asymptomatic COVID-19 patients have the potential to escape the isolation and to contribute to the transmission of the SARS-CoV-2 [20,21].

Our results present peaks of increased cases on study period, the first wave on September 2020, the second wave after the Greek National celebration day on 28 October (week 45), followed from the third wave at weeks 51 and 52. These peaks are similar to the corresponded peak levels reached nationwide at the same period of time. Sport associations or a sport organization like Super League Greece must consider several factors when selecting a testing strategy for a closed community such as elite athletes and staff of football teams. The disease prevalence in the community, and the rate of incidence, will impact the prevalence among the players and staff, and should thus be accounted for. It is note that the vast majority of the identified COVID-19 cases in our study had an asymptomatic COVID-19 infection. Early diagnosis of the infections and appropriate clinical management are important for treating the infected athletes, minimize the risk of transmission, and helped Super League Greece to continue normally the schedule of football matches without any postponement. Another study from Poole et al. supports our results and suggests that continuous testing can help a workplace avoid an outbreak by reducing undetected asymptomatic individuals [22]. Modelling studies estimated that weekly PCR testing to screen workers in health facilities and other high-risk groups, irrespective of symptoms, would reduce their contribution to SARS-CoV-2 transmission by 23% [19]. Promptly tracing of infections is an important measure for the control of COVID-19, aiming to identify and manage contacts of COVID-19 cases in order to reduce further onward transmission. The speed of testing and reporting of results to individuals is critical for isolating cases and initiating contact tracing activities and other public health measures. Minimizing the time between testing and the communication of results will help to maximize the impact of the respective testing strategy and facilitate timely contact management in order to limit ongoing transmission.

The advantages of our study include the prospective cohort nature of the study design, the long period of observation, and the use of a large number of PCR test results among athletes and staff of football teams. Our results are subject to several limitations. We cannot exclude the possibility of information bias in terms of symptoms report. In particular, an underreporting of COVID-19 symptoms may have occurred. In addition, we are unable to report data on the household and social contacts of the cases. Finally, our study was completed in May 2021, six months before the report of omicron variant to World Health Organization from South Africa. Consequently, our results cannot be generalized to the omicron variant. Last, our data did not cover the post vaccination era, the vaccinations of footballers and staff members were started in summer (June 2021), and we were not able to contribute in the current debate related to the use of “green pass” [23]. Future research and preferably cohort studies should investigate the vaccination status of football players and staff of Super League Greece against COVID-19. Previous studies were conducted in elite football players for flu vaccines recorded vaccination coverage ranged between 40% and50%, specifically studied by Signorelli and colleagues in a survey among Italian footballers, who reported an average influenza vaccination rate of 40% [24]. Another study among elite athletes in Super League Greece reported that the seasonal influenza vaccine was recommended by most medical teams (87%), followed by hepatitis B vaccine (62%) and the pneumococcal vaccine (50%) [25].

## 5. Conclusions

In the present study, we found a low incidence of COVID-19 infection among professional football players over a long follow-up period, and no significant transmission events were observed. The majority of the identified COVID-19 cases in our study had an asymptomatic COVID-19 infection. The findings of the present study highlight the importance of appropriate implementing strategies of prevention and surveillance. The choice of testing (RT-PCR) was critical to break transmission chains of severe acute respiratory syndrome coronavirus 2 (SARS-CoV-2) and the coronavirus disease (COVID-19) pandemic in professional football teams.

## Figures and Tables

**Figure 1 vaccines-10-00634-f001:**
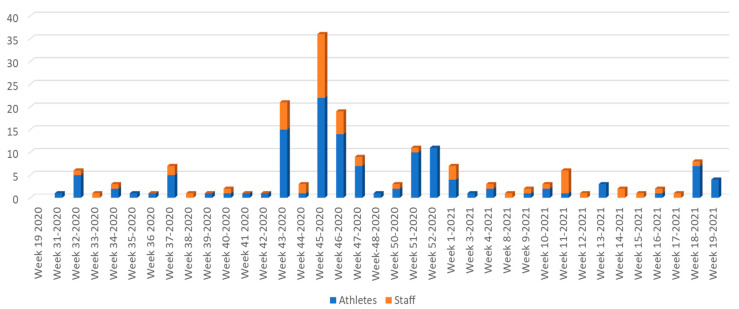
Weekly confirmed cases in athletes and staff in Super League Greece.

**Table 1 vaccines-10-00634-t001:** Data summary.

		Athletes	Staff Members
Test result	Positive	130 (0.57%)	60 (0.27%)
	Negative	22,509 (99.43%)	22,466 (99.73%)
	Total	22,639	22,526
$\hat{R_{1}}$^1^ (95% CI) = 0.57% (0.48%, 0.68%) $\hat{R_{2}}$ ^2^ (95% CI) = 0.27% (0.20%, 0.34%) $\hat{RR}$ ^3^ [(95% CI), *p*-value] = 2.16 ^4^ [(1.59, 2.93), <0.001]			

^1^ risk of a positive test result for athletes, ^2^ risk of a positive test result for staff members ^3^ relative risk (athletes vs. staff), ^4^ 2.16 times higher risk for athletes compared to staff members.

## Data Availability

The data that support the findings of this study are available on request from the corresponding author.

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
