# Peer review of "COVID-19 Infection among Elite Football Players: A Nationwide Prospective Cohort Study"

_vaccines, 2022, doi:10.3390/vaccines10050634_

Round 1
Reviewer 1 Report
the manuscript aims to explore the incidence of covid-19 among professional football players
Yes, it is relevant and original since COVID-19 is still an important health challenge and no previous longitudinal studies have been conducted so far on this topic
The longitudinal perspective is really relevant for the aim of the study
The method is appropriate and clearly reported, however, a new section explaining variables used is missing. please add
In the results section: the manuscript will greatly benefit if the authors will add a descriptive characteristics table of the included subjects.
In the discussion, please make your results in light of previous researches. As examples, consider to refer to:
1) Krzywański J, Mikulski T, Krysztofiak H, et al. Elite athletes with COVID-19 - Predictors of the course of disease. J Sci Med Sport. 2022;25(1):9-14. doi:10.1016/j.jsams.2021.07.003
2) Gianfredi V, Mauer NS, Gentile L, Riccò M, Odone A, Signorelli C. COVID-19 and Recreational Skiing: Results of a Rapid Systematic Review and Possible Preventive Measures. Int J Environ Res Public Health. 2021 Apr 20;18(8):4349. doi: 10.3390/ijerph18084349
3) Da Silva et al. Athletes health during pandemic times: hospitalization rates and vriables related to COVID-19 Prevalence among Endurance Athletes. Int J Cardiovasc Sci 34 (3) • May-Jun 2021 https://doi.org/10.36660/ijcs.20200208
Please, consider to add a graphical abstract. usually, it improves the interest of readers and readability of the manuscript
Author Response
The manuscript aims to explore the incidence of covid-19 among professional football players. Yes, it is relevant and original since COVID-19 is still an important health challenge and no previous longitudinal studies have been conducted so far on this topic. The longitudinal perspective is really relevant for the aim of the study. The method is appropriate and clearly reported, however, a new section explaining variables used is missing. please add
We would like to thank the reviewer for the constructive comments to help us to improve the present manuscript. It is clear from our statistical analysis subsection that our dependent variable is the incidence of COVID -19 infection among footballers and staff of football teams.
In the results section: the manuscript will greatly benefit if the authors will add a descriptive characteristics table of the included subjects.
Response: We would like to thank the reviewer for the comment. We have included data related to the demographic characteristics of the participants in the methods section of the revised form (Lines 88-89).
In the discussion, please make your results in light of previous researches. As examples, consider to refer to:
1) Krzywański J, Mikulski T, Krysztofiak H, et al. Elite athletes with COVID-19 - Predictors of the course of disease. J Sci Med Sport. 2022;25(1):9-14. doi: 10.1016/j.jsams.2021.07.003
2) Gianfredi V, Mauer NS, Gentile L, Riccò M, Odone A, Signorelli C. COVID-19 and Recreational Skiing: Results of a Rapid Systematic Review and Possible Preventive Measures. Int J Environ Res Public Health. 2021 Apr 20;18(8):4349. doi: 10.3390/ijerph18084349
3) Da Silva et al. Athletes health during pandemic times: hospitalization rates and variables related to COVID-19 Prevalence among Endurance Athletes. Int J Cardiovasc Sci 34 (3) • May-Jun 2021 https://doi.org/10.36660/ijcs.20200208
Response: This reference along with reference 3 (Da Silva et al) were included in the introduction and reports on the mild disease status among COVID 19 infected footballers (line 71). The 3rd reference was added in the discussion section (line 168). New references (5,6,17).
Please, consider to add a graphical abstract. usually, it improves the interest of readers and readability of the manuscript
Response: We would like to thank the reviewer for this comment. We are at Editorial Office’s disposal for any further clarification.

Reviewer 2 Report
The reviewed manuscript title is “Risk of COVID -19 in professional football players of Super League in Greece: a prospective cohort study”, a prospective epidemiological and laboratory study to determine the risk factors for Covid-19 among the football players and staff members in Greece.
Abstract part
The authors should use the term of “COVID-19” along the text body.
Line 22: “…Reverse Transcriptase Polymerase chain reaction (RT-PCR)..” can be revised as “..reverse transcriptase-polymerase chain reaction (RT-PCR) test for the detection of SARS CoV2” or “real time RT-PCR test for the detection of SARS CoV2”.
Line 24: The sentence should be revised as “Out of the 190 cases which were turned positive, 64 (34%) of cases were considered as symptomatic and 126 (66%) cases were asymptomatic.”
Line 26: The sentence should be revised as “The incidence rate of a positive test result for footballers was 0.57%, (Confidence Interval [CI] 0.48% - 0.68%) and for staff members was 0.27%, (CI 0.20% - 0. 34%, respectively.
Line 28: “95%CI= 1.59, 2.93” should be revised as “95%CI 28 1.59 - 2.93”.
Introduction part
Line 38: The reference no 1 should be removed.
Materials and Methods part
Since this study is a prospective character, it should be disclose for the epidemiological (i.e. age, gender), some of the associated risk factors (diabetes mellitus, hypertension etc.) for staff members and COVID-19 vaccination status should be investigated. Further, which symptoms were present among the symptomatic cases and the median number of COVID-19 tests should also be investigated.
Results part
The Figure 1 and Table 2 have no make sense and they should be deleted from the manuscript.
Author Response
The reviewed manuscript title is “Risk of COVID -19 in professional football players of Super League in Greece: a prospective cohort study”, a prospective epidemiological and laboratory study to determine the risk factors for Covid-19 among the football players and staff members in Greece.
Abstract part
The authors should use the term of “COVID-19” along the text body.
Response: We would like to thanks for your comment. We have modified the text, accordingly.
Line 22: “…Reverse Transcriptase Polymerase chain reaction (RT-PCR)..” can be revised as “..reverse transcriptase-polymerase chain reaction (RT-PCR) test for the detection of SARS CoV2” or “real time RT-PCR test for the detection of SARS CoV2”.
Response: We have modified the abstract, accordingly.
Line 24: The sentence should be revised as “Out of the 190 cases which were turned positive, 64 (34%) of cases were considered as symptomatic and 126 (66%) cases were asymptomatic.”
Response: We would like to thanks for your comment. We have modified the abstract, accordingly
Line 26: The sentence should be revised as “The incidence rate of a positive test result for footballers was 0.57%, (Confidence Interval [CI] 0.48% - 0.68%) and for staff members was 0.27%, (CI 0.20% - 0. 34%, respectively.
Response: Done.
Line 28: “95%CI= 1.59, 2.93” should be revised as “95%CI 28 1.59 - 2.93”.
Response: Done.
Introduction part
Line 38: The reference no 1 should be removed.
Response: This reference has been removed.
Materials and Methods part
Since this study is a prospective character, it should be disclose for the epidemiological (i.e. age, gender), some of the associated risk factors (diabetes mellitus, hypertension etc.) for staff members and COVID-19 vaccination status should be investigated. Further, which symptoms were present among the symptomatic cases and the median number of COVID-19 tests should also be investigated.
Response: We have revised the text, accordingly and we have included on the revised text data on the sex and age distribution among participants (lines 88-89). We are not able to report more information’s with respect of symptoms. This is also the case for the associate factors (diabetes mellitus, hypertension etc.) these data were not available.

Reviewer 3 Report
The aim of our study was to investigate the incidence, Relative Risk and characteristics of COVID 19 infection among high level footballers. This is a very great study which aI have a minor revisions prior the acceptance for publication.
-Introduction: Please, to remove "According to the protocol, before the start of the games, all the players, the match officials, the coaching staff and team staff will need to be re-examined for COVID 19. Prerequisite for the start of the games, is the examination and especially the real time RT PCR molecular test. Depending on the test results, in case of a positive case the instructions of the competent bodies for health issues are followed, while in case of a negative test result the subject will be able to take part in the football game. The medical team of each football team should have clinically examined all footballers and coaching staff and get detailed history (with emphasis on clinical signs and symptoms associated with recent COVID 19 infection (both for him as well as for his family and social environment). In case of a laboratory confirmed COVID-19 infection, the athlete will be referred for extensive clinical examination and laboratory testing. The footballers’ medical records should be kept by the medical team of each football club under the terms and legislation for personal data protection [6]. Due to the nature of their job (e.g. direct contact with others) this occupational group has a theoretical risk of infectious diseases including COVID 19."
-Methods: What is sample size calculus? were these subjects recruited by convenience?
-Results: Were the data to masks asked?
-Discussion: In the first paragraph, second sentence, please to add a reference for "This rate is very similar to that reported by Pedersen et al among Danish elite footballers."
Author Response
Reviewer 3.
The aim of our study was to investigate the incidence, Relative Risk and characteristics of COVID 19 infection among high level footballers. This is a very great study which aI have a minor revision prior the acceptance for publication.
-Introduction: Please, to remove "According to the protocol, before the start of the games, all the players, the match officials, the coaching staff and team staff will need to be re-examined for COVID 19. Prerequisite for the start of the games, is the examination and especially the real time RT PCR molecular test. Depending on the test results, in case of a positive case the instructions of the competent bodies for health issues are followed, while in case of a negative test result the subject will be able to take part in the football game. The medical team of each football team should have clinically examined all footballers and coaching staff and get detailed history (with emphasis on clinical signs and symptoms associated with recent COVID 19 infection (both for him as well as for his family and social environment). In case of a laboratory confirmed COVID-19 infection, the athlete will be referred for extensive clinical examination and laboratory testing. The footballers’ medical records should be kept by the medical team of each football club under the terms and legislation for personal data protection [6]. Due to the nature of their job (e.g. direct contact with others) this occupational group has a theoretical risk of infectious diseases including COVID 19."
Response: We have modified the text accordingly.
-Methods: What is sample size calculus? were these subjects recruited by convenience?
Response: All football players and staff members of Super League Greece were recruited to take part in the study. Response rate 100%.
-Results: Were the data to masks asked?
Response: We didn’t record data about the masks. According to the hygiene protocol all athletes and staff were required to wear masks upon arrival at the sports facility.
-Discussion: In the first paragraph, second sentence, please to add a reference for "This rate is very similar to that reported by Pedersen et al among Danish elite footballers."
Response: Thanks for the comment. We refer to the number [9] reference Petersen et al. line 147.

Round 2
Reviewer 2 Report
Dear authors,
Thank you very much for made revisions by you.
Sincerely